# Geometric Regularity with Robot Intrinsic Symmetry in Reinforcement Learning

Shengchao Yan, Yuan Zhang, Baohe Zhang, Joschka Boedecker, Wolfram Burgard[†]
Department of Computer Science, University of Freiburg, Germany
[†]Department of Engineering, University of Technology Nuremberg, Germany

*Abstract*—Geometric regularity, which leverages data symmetry, has been successfully incorporated into deep learning architectures such as CNNs, RNNs, GNNs, and Transformers. While this concept has been widely applied in robotics to address the curse of dimensionality when learning from high-dimensional data, the inherent reflectional and rotational symmetry of robot structures has not been adequately explored. Drawing inspiration from cooperative multi-agent reinforcement learning, we introduce novel network structures for deep learning algorithms that explicitly capture this geometric regularity. Moreover, we investigate the relationship between the geometric prior and the concept of Parameter Sharing in multi-agent reinforcement learning. Through experiments conducted on various challenging continuous control tasks, we demonstrate the significant potential of the proposed geometric regularity in enhancing robot learning capabilities.

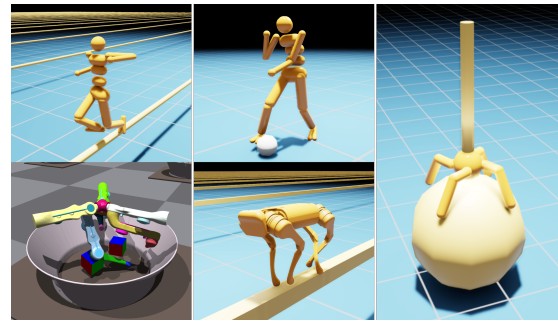

Fig. 1: We design tasks (except TriFinger [3]) challenging for current deep reinforcement learning baseline algorithms.

## I. Introduction

Robots have the ability to undertake tasks that are dangerous or difficult for humans. With more degrees of freedom, they can perform increasingly complex tasks. For example, humanoid robots and quadrupedal robots can walk over challenging terrain, while robot arms and hands can achieve dexterous manipulation. However, controlling robots with a large number of degrees of freedom becomes increasingly difficult as the observation and action space grows exponentially. Although deep reinforcement learning has been employed to solve various robot control problems [8, 11, 20, 3], learning effective control strategies for these robots remains a challenging task.

Training neural networks on high-dimensional data is known to be challenging due to the curse of dimensionality [4]. To overcome this challenge, researchers have developed network architectures and incorporated various inductive biases that respect the structure and symmetries of the corresponding domains. For example, convolutional neural networks (CNNs) leverage the strong geometric prior of images by incorporating translation equivariance into the design of convolutional layers. This ensures that the extracted features move along with the original image, regardless of the direction it is shifted in. Similarly, graph neural networks (GNNs) take advantage of the geometric prior of permutation invariance in other domains to capture the relationships among objects. Overall, incorporating domain-specific inductive biases and symmetries can greatly improve the ability of neural networks to learn from high-dimensional data.

However, in the realm of deep reinforcement learning research, the potential benefits of utilizing symmetry structures present in environments, such as reflectional and rotational symmetry, have not attracted much attention and thus, how to combine these prior knowledge to effectively improve the existing approaches still is worth to be investigated. To bridge the research gap, we propose to reformulate the control problems under Multi-Agent Reinforcement Learning (MARL) framework to better leverage the symmetry structures. We demonstrate the surprising effectiveness of our approach by combining the new architectures with model-free deep reinforcement learning methods. Additionally, we establish a connection between our proposed geometric prior and the important concept of "Parameter Sharing" in multi-agent reinforcement learning, which excessively reduces the optimization space and speeds up the learning process. We also design a set of challenging robot control tasks (see Fig. 1) and evaluate our method on them. Our experimental results show that our proposed method significantly improves the performance of robot control learning tasks.

## II. Background and Related Work

### A. Multi-Agent Reinforcement Learning (MARL)

MARL is an extended reinforcement learning method for decision-making problems, where multiple agents can interact and learn in one environment. The most popular mathematical framework for MARL problems is Markov games. A Markov game is a tuple $\langle \mathcal{N}, \mathcal{S}, \mathcal{O}, \mathcal{A}, P, R_i, \gamma \rangle$. $\mathcal{N}$ is the set of all agents and $\mathcal{S}$ is the set of states. $\mathcal{O}_i$ and $\mathcal{A}_i$ are observation space and action space for agent $i$, while $\mathcal{O} = \times_{i \in \mathcal{N}} \mathcal{O}_i$ and $\mathcal{A} = \times_{i \in \mathcal{N}} \mathcal{A}_i$ represent joint observation space and joint action space. Define $\Delta_{|S|}$ and $\Delta_{|A|}$ be the probability measure on $\mathcal{S}$ and $\mathcal{A}$ respectively. Then $P$ is the transition

probability $P(s'|s,a) : \mathcal{S} \times \mathcal{A} \to \Delta_\mathcal{S}$. Each agent $i$ maintains a specific reward function $R_i(s,a) : \mathcal{S} \times \mathcal{A} \to \mathbb{R}$, and the future rewards are discounted by the discount factor $\gamma \in [0,1]$. Let $\Pi_i = \{\pi_i(a_i|o_i) : \mathcal{O}_i \to \Delta_{\mathcal{A}_i}\}$ be the policy space for agent $i$, then the objective for agent $i$ is represented as $\max_{\pi_i} \mathbb{E}_{\pi,P} \left[ \sum_{t=0}^{+\infty} \gamma^t R_i(s_t, a_t) \right]$. In practice, the state space and the observation space can be identical if the observation has already fully described the system. Our paper also follows this assumption and hence uses observation alone.

Multi-Agent Mujoco [13] is a popular benchmark for MARL algorithms which divides a single robot into several distinct parts with separate action space. However, the state-of-the-art MARL algorithms still couldn't match the performance of the single-agent algorithms on this benchmark. Different from their work, in which they arbitrarily divide robots into parts and ignore the geometric structures of the robots, we leverage ideas from geometric regularity during the MARL training and our results show that MARL can outperform single-agent algorithms by a substantial margin.

### B. Symmetry in Robot Learning

In robot learning domain, two groups of symmetric structures have been used to improve performance and learning efficiency. 1) **Extrinsic Symmetry**: By extrinsic symmetry we refer to the symmetries existing in the Exteroceptive sensors of the robot such as camera input. Some work [18, 24, 17, 19] have been proposed to integrate these symmetries into system identification via the neural network, especially CNN-structured network. These methods can largely improve the performance for manipulation tasks, but they are mostly around manipulation tasks with image input and gripper without roll-pitch movement. Van der Pol et al. [16] introduce MDP homomorphic networks to numerically construct equivariant network layers.However, the proposed network only considers a pole balancing task with discrete action. Moreover, additional calculation is required to design the network even if the domain specific transformation is given. Mondal et al. [12] propose to learn symmetry directly from data in the latent space but is still limited to representation learning from images. 2) **Intrinsic Symmetry**: Different from extrinsic symmetries, intrinsic symmetries mostly naturally come from the physical constraints in the control system. For example, a humanoid robot control task exhibits reflectional symmetry. A symmetric control policy on such robot is usually more natural and effective. Mavalankar [10] proposes a data-augmentation method to improve reinforcement learning method for rotation invariant locomotion. Abdolhosseini et al. [2] investigate four different methods to encourage symmetric motion of bipedal simulated robots. They are implemented via specific policy network, data augmentation or auxiliary loss function. Even though the robots' motions become more natural-looking, they do not show a major improvement on different tasks. The policy network method in [2] is similar to ours in this work. But instead of a specific network merely for locomotion tasks with reflectional symmetry, we propose a generic equivariant policy network for both reflectional and rotational symmetries,

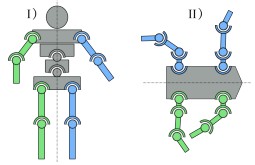 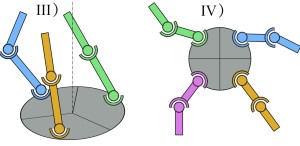

(a) Reflectional symmetry      (b) Rotational symmetry

Fig. 2: **Agent partitioning considering symmetry structures**: Humanoid and Cheetah robots split into left and right parts by reflectional symmetry; TriFinger and Ant robots split into 3 and 4 parts by rotational symmetry, where each part is controlled individually by a dedicated agent. The central part (grey) is controlled by all agents.

which are predominant symmetry features in robotic systems and animal biology. Moreover, we approach the control task in the field of multi-agent systems. Finally, we get substantial performance improvement in experiments by reducing the policy search space.

### III. SINGLE ROBOT CONTROL AS MARL

Instead of learning a single-agent policy to control the whole robot, which will lead to a large observation-action space that is difficult to optimize, we introduce multiple agents that are responsible for each individual component of the robot inspired by MARL. We further propose a framework driven by the presence of symmetry structures in many robots and exploit such inductive biases to facilitate the training by applying parameter sharing techniques.

The overview structure of our method is to (1) identify the geometric structures of different robots and divide single robots into multiple parts accordingly; (2) reformulate the control problem as a MARL framework; (3) optimize policies with parameter sharing technique.

### A. Dividing Single Robots into Multiple Parts

Previous research [13] also divides a single robot into multiple parts to evaluate the performance of MARL methods. However, its irregular partitioning makes the multi-agent methods hard to compete with the single-agent methods. In this paper, we reconsider partitioning in a more reasonable way, which is achieved by taking into account the symmetry structures of robots when dividing them into multiple agents.

As shown in Fig. 2a, robots with reflectional symmetry can be partitioned into left (blue), right (green) and central (grey) parts. The robots with rotational symmetry in Fig. 2b are partitioned into parts with the same number of symmetric limbs (colour) and a central part (grey). For a robot with any of these symmetric structures, we split the whole robot's original observation-action space $\mathcal{O} \times \mathcal{A}$ by $\mathcal{O} = \mathcal{O}_c \times \prod_{i \in \mathcal{N}} \mathcal{O}_{s,i}$ and $\mathcal{A} = \mathcal{A}_c \times \prod_{i \in \mathcal{N}} \mathcal{A}_{s,i}$. $\mathcal{O}_c \times \mathcal{A}_c$ represents the central observation-action pair, which consists of measurements and actuators that do not have symmetric counterparts, such as the position, orientation, velocity and joints of the torso, target direction, or states of the manipulated objects. Raw

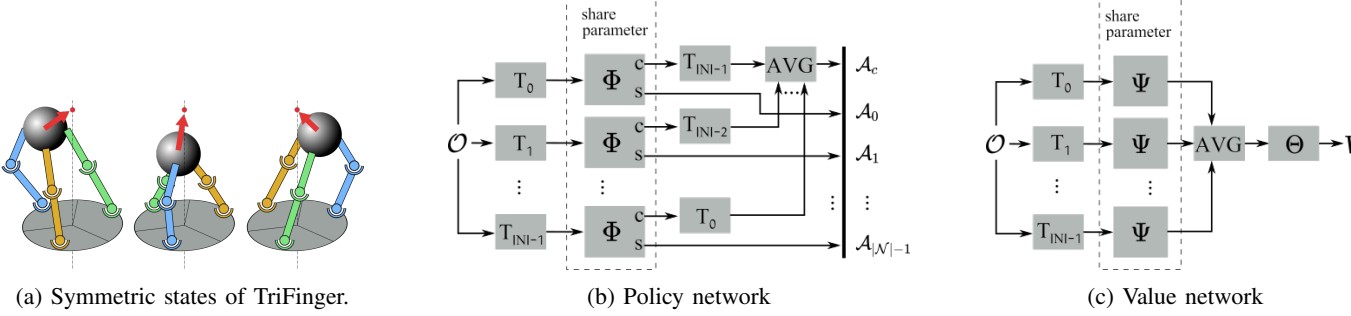

(a) Symmetric states of TriFinger.   (b) Policy network   (c) Value network

Fig. 3: a) TriFinger robot moves a sphere towards a target position. From left to right are the original state, rotated by $120°$, and rotated by $240°$. Note that the actions of different body parts should be equivariant with regard to the transformation. The red arrow represents the desired moving direction of the manipulated object. b) Equivariant policy network with parameter $\Phi$. **c** and **s** stand for central and symmetric actions. c) Invariant value network with parameter $\Psi, \Theta$.

sensor data such as images and point clouds also belongs to central observation. $\mathcal{O}_{s,i} \times \mathcal{A}_{s,i}$ corresponds to symmetric observation-action spaces, whose measurements may include joint positions and velocities from the limbs, contact sensor measurements of the feet or fingers, and so on. The symmetric observation-action spaces are exactly the same for any $i \in \mathcal{N}$ due to the robots' symmetric property.

### B. Multi-agent Reinforcement Learning Formulation

Assume the original observation and action of the whole robot be $o \in \mathcal{O}$ and $a \in \mathcal{A}$ respectively and the number of agents $|\mathcal{N}|$, equal to the number of symmetry parts of the robots. For each agent $i \in \mathcal{N}$, there is a unique transformation function $T_i$ to obtain its own observation $o_i = T_i(o)$. Detailed explanation of $T_i$ can be found in Appendix A1. Each agent generates the local action $a_i$, consisting of $a_{c,i} \in \mathcal{A}_c$ and $a_{s,i} \in \mathcal{A}_{s,i}$ for central and symmetric actions, by its own policy network. Finally, the whole robot's action $a$ is recovered by gathering all symmetric actions $a_{s,i}$ and merging all central actions $a_{c,i}$ into $a_c$.

Regarding the reward function, our formulation follows the cooperative MARL setup, where $R_i$ for all $i \in \mathcal{N}$ are identical at every time step. This shared reward is calculated by a task-related reward function $R(o, a)$ which depends on the whole robot's observation and action. To optimize the policies $\pi_i$, we adopt the multi-agent version of Proximal Policy Optimization (PPO) [14] methods. PPO is a popular model-free actor-critic reinforcement learning algorithm in different domains [22, 3, 11] for its stability, good performance and ease of implementation. Its multi-agent version also achieves competitive performance on different MARL benchmarks [23, 6].

### C. Geometric Regularization

*Parameter Sharing* has been recognized as a crucial element in MARL for efficient training [7]. By enabling agents to share parameters in their policy networks, parameter sharing not only facilitates scalability to a large number of agents but also enables agents to leverage shared learned representations, leading to reduced training time and improved overall performance. However, it is shown by Christianos et al.

[5] that indiscriminately applying parameter sharing could hurt the learning process. Successful utilization of parameter sharing relies on the presence of homogeneous agents as a vital requirement. In other words, agents should execute the same action once they are given the same observation. This assumption ensures the transformation equivariance of the overall policy regarding the symmetry structures.

Take the simplified TriFinger Move task as an example, where the TriFinger robot has to move the sphere towards a target position. As shown in Fig. 3a, if the whole system is rotated by $120°$ or $240°$ around the $z$ axis of the robot base, the actions should also shift circularly among the three fingers for the optimal policy. Given the whole robot's observation $o$, this relationship can be denoted by:

$$A_{s,j}(T_i(o)) = A_{s,i}(T_j(o)), \quad A_c(T_i(o)) = T_i(A_c(o)) \quad (1)$$

where $A_{s,j}$ is the symmetric action of the $j$th agent, $A_c$ is the central action, $T_i$ is the symmetry transformation between agents $i$ and $0$ (see definition in Appendix A1). The transformation for observation and action are so similar that we won't distinguish between them in this work for simplicity. Note that the the corresponding robot parts of agents can be defined arbitrarily. It does not influence the equivariance/invariance.

Based on the equivariance represented by Eq. 1, we design the multi-agent actor-critic network structure in Fig. 3b, 3c. Agent $i$ gets a transformed observation $T_i(o)$ as the input of the policy network, the output action value consists of $a_{c,i}$ and $a_{s,i}$. The central joints are controlled by the mean action over all agents' output $a_{c,i}$, while $a_{s,i}$ will be used as the action to take for the robot part $i$. The policy network parameters are shared among agents. The value network gets the observations from all agents as input. The observations first go through the shared feature learning layers in the value network. Then the latent features are merged by a set operator (*mean* in this work). The value is finally calculated with the merged feature.

The proposed policy network is equivariant with respect to symmetric transformations we consider in this work, while the value network is an invariant function (see proof in Appendix A2). By sharing the same policy network among all

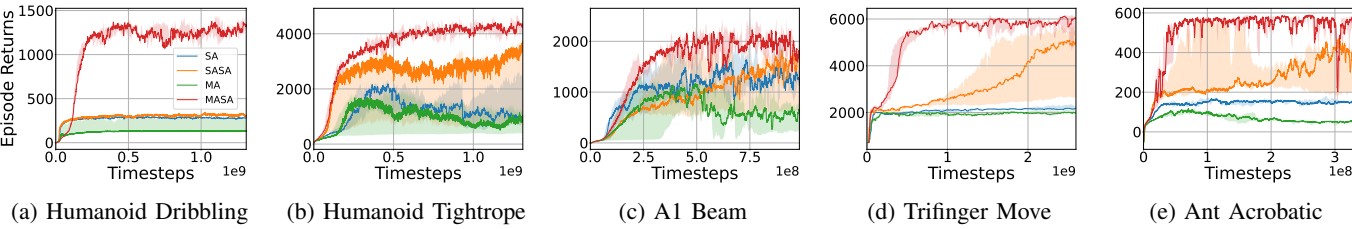

| (a) Humanoid Dribbling | (b) Humanoid Tightrope | (c) A1 Beam | (d) Trifinger Move | (e) Ant Acrobatic |

Fig. 4: Learning curves on robot control tasks. The x-axis is environment time steps and the y-axis is episodic returns during training. All graphs are plotted with median and 25%-75% percentile shading across 5 random seeds.

agents, we are able to incorporate the geometric regularization and reduce the dimension of the observation-action space.

## IV. EXPERIMENTS AND DISCUSSION

### A. Experimental Setup

*1) Challenging Tasks:* Previous robotic control benchmarks [15] evaluate algorithms on fundamental tasks, e.g. controlling agents to walk. The movements in these tasks are limited and it's relatively easy to learn an optimal policy. In this work, we design several more challenging robotic control tasks, where current state-of-the-art methods fail to achieve good performance. The tasks are shown in Fig. 1: Humanoid Tightrope, Humanoid Dribbling, A1 Beam, Trifinger Move and Ant Acrobatic. The detailed introduction of the tasks can be found in Appendix B2. All experiments are carried out based on the NVIDIA Isaac Gym [9] robotics simulator.

*2) Baselines:* For each task, we compare our method, named as Multi-agent with Symmetry Augmentation (*MASA*), with a set of baselines including:

- Single-agent (*SA*): We first compare the single-agent reinforcement learning algorithm, which optimize all of the robot parts jointly. This baseline can provide an intuitive comparison of our proposed framework to previous classic reinforcement learning works. The state space is kept the same as the multi-agent one for a fair comparison.
- Single-agent with Symmetry Augmentation (*SASA*): This baseline follows the *SA*'s setup and is augmented with a symmetry loss [2]. Specifically, for any received observation $o$, we calculate its symmetric representation $T_i(o)$. We regulate the policy function $\pi$ and the value function $V$ in PPO with extra symmetry losses by minimizing $\|T_i(A(o)) - A(T_i(o))\|_2$ and $|V(o) - V(T_i(o))|$, where $A$ and $V$ are the gathered action and critic value of the robot.
- Multi-agent without Symmetry Augmentation (*MA*): This baseline uses the same architecture as *MASA*. However, it does not involve the transformations in Fig. 3b 3c. Thus the geometric regularity of symmetry is ignored, which follows the previous research [13]. We concatenate a one-hot id encoding to each agent's observation as a common operation for non-homogeneous agents.

We conclude the hyperparameters in Appendix B1.

### B. Main Results

Figure 4 presents the average return of all methods on different tasks during training. The proposed method *MASA*

significantly outperforms other baselines across all 5 tasks. Further, the advantages over other baselines rise with the increasing difficulties of the task, which can be indicated by the increased number of joints, the extended state dimension and the enlarged state space in the task. Humanoid Tightrope and Humanoid Football control the same robot. However, in the tightrope task, the robot only needs to walk forward, while the football task involves random turns and manipulating an external object, so that other baselines can hardly learn meaningful behaviours in this task.

By comparing the results of *MASA*, *MA* and *SASA*, we could observe that both of the two factors in *MASA*, multi-agent framework and symmetry structure, play an important role. Utilizing symmetry data structure alone (*SASA*) can gradually learn to solve a few tasks but with aparently lower data efficiency. Because the optimization space is not reduced and thus larger than that of *MASA* method. The multi-agent structure itself (*MA*) cannot guarantee meaningful results at all, which follows the criticism of naively sharing parameters among non-homogeneous agents [5].

In the Humanoid Dribbling task, *MASA* initially underperforms compared to other baselines. This is because the baseline methods prioritize self-preservation and struggle to find a policy that balances dribbling and staying alive. By focusing on avoiding falling down and kicking the ball too far away, they learn to stand still near the ball while disregarding the rewards associated with ball movement. Consequently, the baseline agents are able to survive longer at the beginning, resulting in higher returns compared to *MASA*.

### C. Discussion

Our proposed multi-agent method exhibits impressive performance in challenging control tasks. The network structures we introduce are not limited to on-policy reinforcement learning algorithms and can be adapted for off-policy learning, imitation learning, and model-based learning methods. While our approach is straightforward to implement with observation transformations, it still requires domain knowledge. We believe our method can enhance robot learning in more demanding tasks, serving as a guide for designing robots with increased degrees of freedom while managing the observation-action space growth linearly. Future research directions include exploring additional symmetric structures and automating the process of identifying robots' intrinsic symmetries.

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

## APPENDIX

### A. Extra Method Details

*1) Transformation Functions:* As mentioned in Sec. III-C, $T_0$ of the base agent is identity transformation. In this section we describe in detail the transformation function of other agents. For convenience, we only explain observation transformation in detail, the transformations are actually the same for actions, for which we only need to change the observation to the corresponding action components. By default, we assume the original observation $o = [o_c, o_{s,0}, o_{s,1}, \ldots, o_{s,|\mathcal{N}|-1}]$ is in the local coordinate system of the robot base for convenience.

*a) Reflectional Symmetry:* For robots with reflectional symmetry, two robot parts in Fig. 2a are controlled by agents $\{0, 1\}$. We define $T_1(o) = [T_{c,1}(o_c), o_{s,1}, o_{s,0}]$, where $T_{c,1}(o_c)$ is a reflectional function, which reflects the central observation through the plane of symmetry. As a result of $T_1(o)$ different observation components are transformed as follows:

- symmetric observations directly switch their values;
- some of the central observation values are negated;
  - humanoid robot: $y_{torso}$, $v_{torso,y}$, $\omega_{torso,x}$, $\omega_{torso,z}$, $\alpha_{torso}$, $\gamma_{torso}$, $\theta_{lower\ waist,x}$, $\theta_{pelvis,x}$, $\omega_{lower\ waist,x}$, $\omega_{pelvis,x}$, $a_{lower\ waist,x}$, $a_{pelvis,x}$
  - A1 robot: $y_{torso}$, $v_{torso,y}$, $\omega_{torso,x}$, $\omega_{torso,z}$, $\alpha_{torso}$, $\gamma_{torso}$
  - external objects: $y_{ball}$, $v_{ball,y}$
- other central observation values stay the same.

*b) Rotational Symmetry:* For robots with rotational symmetry, the robot parts in Fig. 2b are controlled by agents $\{0, 1, \ldots, |\mathcal{N}| - 1\}$. We define $T_i(o) = [T_{c,i}(o_c), o_{s,i}, o_{s,i+1}, \ldots, o_{s,|\mathcal{N}|-1}, o_{s,0}, o_{s,1}, \ldots, o_{s,i-1}]$, where $T_{c,i}(o_c)$ is a rotational transformation for central observations around the axis of symmetry. The degree of rotation is the angular distance from the robot part of agent $i$ to that of agent 0 along the axis of symmetry. Taking the TriFinger robot in Fig. 3 as an example, the rotation angles are $0, 120$ and $240$ degrees for the three agents. As a result of $T_i(o)$ different observation components are transformed as follows,

- symmetric observations circularly shift their values;
- the central observation components are rotated.

*2) Proof of Transformation Equivariance/Invariance:* At the beginning we summarize the properties of the symmetry transformations in this work. They are:

- commutative: $T_j(T_i(o)) = T_{i+j}(o) = T_i(T_j(o))$
- distributive: $T_j(T_i(o) + T_k(o)) = T_j(T_i(o)) + T_j(T_k(o))$
- cyclic: $T_i(o) = T_{i+|\mathcal{N}|}(o)$

The equivariance of the policy for symmetric actions in Eq. 1 is proved as follows:

$$A_{s,j}(T_i(o)) = \Phi_s(T_j(T_i(o))) = \Phi_s(T_j(T_i(o)))$$
$$= A_{s,i}(T_j(o))$$

The equivariance for the central action is proved as follows:

$$A_c(T_i(o)) = \frac{1}{|\mathcal{N}|} \sum_{j=0}^{|\mathcal{N}|-1} T_{|\mathcal{N}|-1-j}(\Phi_c(T_j(T_i(o))))$$

$$= \frac{1}{|\mathcal{N}|} \sum_{j=|\mathcal{N}|-i}^{2|\mathcal{N}|-i-1} T_{|\mathcal{N}|-1-j}(\Phi_c(T_{i+j}(o)))$$

$$= \frac{1}{|\mathcal{N}|} \sum_{k=|\mathcal{N}|}^{2|\mathcal{N}|-1} T_{|\mathcal{N}|+i-1-k}(\Phi_c(T_k(o)))$$

$$= \frac{1}{|\mathcal{N}|} \sum_{k=0}^{|\mathcal{N}|-1} T_i(T_{|\mathcal{N}|-1-k}(\Phi_c(T_k(o))))$$

$$= T_i\left(\frac{1}{|\mathcal{N}|} \sum_{k=0}^{|\mathcal{N}|-1} T_{|\mathcal{N}|-1-k}(\Phi_c(T_k(o)))\right)$$

$$= T_i(A_c(o))$$

The invariance of the value network is proved as follows:

$$V(T_i(o)) = \Theta\left(\frac{1}{|\mathcal{N}|} \sum_{j=0}^{|\mathcal{N}|-1} \Psi(T_j(T_i(o)))\right)$$

$$= \Theta\left(\frac{1}{|\mathcal{N}|} \sum_{j=|\mathcal{N}|-i}^{2|\mathcal{N}|-i-1} \Psi(T_{i+j}(o))\right)$$

$$= \Theta\left(\frac{1}{|\mathcal{N}|} \sum_{k=0}^{|\mathcal{N}|-1} \Psi(T_k(o))\right)$$

$$= V(o) = V(T_j(o))$$

### B. Extra Experimental Setups

*1) Hyperparameters:* Each baseline is run with 5 random seeds. All experiments are carried out on GPU card NVIDIA A100 and rtx3080 GPU. The hyperparameters of all baselines are consistent for a fair comparison. The detailed values can be accessed in Table I.

*2) Tasks Details:*

*a) Humanoid Tightrope:* In this task, the agent learns to control a humanoid robot to walk on a tightrope. The humanoid robot has 21 controllable motors. The tightrope is extremely narrow with a diameter of only $10\,\mathrm{cm}$, which challenges the efficiency of learning algorithms. The agent is rewarded with a forward speed on the tightrope and a proper posture. At each non-terminating step, the reward $r = w_v \times r_v + w_{alive} \times r_{alive} + w_{up} \times r_{up} + w_{heading} \times r_{heading} + w_{action} \times r_{action} + w_{energy} \times r_{energy} + w_{lateral} \times r_{lateral}$, where

- $r_v$ is the robot's forward velocity, $w_v = 1.0$;
- $r_{alive} = 1$, $w_{alive} = 2.0$;
- $r_{up} = 1$ if $e_{up,z} > 0.93$, where $e_{up}$ is the basis vector of torso's $z$ axis in the global coordinate system, otherwise the value is 0, $w_{up} = 0.1$;
- $r_{heading} = e_{forward,x}$, where $e_{forward}$ is the basis vector of torso's $x$ axis in global coordinate system, $w_{forward} = 0.1$;
- $r_{action} = \|a\|_2^2$, where $a$ is joints action, $w_{action} = -0.01$
- $r_{energy}$ is the joints power consumption, $w_{energy} = -0.05$

TABLE I: Hyperparameters of all experiments.

| HYPERPARAMETERS | HUMANOID TIGHTROPE | HUMANOID FOOTBALL | TRIFINGER MOVE | A1 BEAM | ANT ACROBATIC |
|---|---|---|---|---|---|
| BATCH SIZE | 4096×32 | 4096×32 | 16384×16 | 4096×24 | 4096×16 |
| MIXED PRECISION | TRUE | TRUE | FALSE | TRUE | TRUE |
| NORMALIZE INPUT | TRUE | TRUE | TRUE | TRUE | TRUE |
| NORMALIZE VALUE | TRUE | TRUE | TRUE | TRUE | TRUE |
| VALUE BOOTSTRAP | TRUE | TRUE | TRUE | TRUE | TRUE |
| NUM ACTORS | 4096 | 4096 | 16384 | 4096 | 4096 |
| NORMALIZE ADVANTAGE | TRUE | TRUE | TRUE | TRUE | TRUE |
| GAMMA | 0.99 | 0.99 | 0.99 | 0.99 | 0.99 |
| GAMMA | 0.95 | 0.95 | 0.95 | 0.95 | 0.95 |
| E-CLIP | 0.2 | 0.2 | 0.2 | 0.2 | 0.2 |
| ENTROPY COEFFICIENT | 0.0 | 0.0 | 0.0 | 0.0 | 0.0 |
| LEARNING RATE | 5.E-4 | 5.E-4 | 3.E-4 | 3.E-4 | 3.E-4 |
| KL THRESHOLD | 0.0008 | 0.0008 | 0.0008 | 0.0008 | 0.0008 |
| TRUNCATED GRAD NORM | 1.0 | 1.0 | 1.0 | 1.0 | 1.0 |
| HORIZON LENGTH | 32 | 32 | 16 | 24 | 16 |
| MINIBATCH SIZE | 32768 | 32768 | 16384 | 32768 | 32768 |
| MINI EPOCHS | 5 | 5 | 4 | 5 | 4 |
| CRITIC COEFFICIENT | 4.0 | 4.0 | 4.0 | 2.0 | 2.0 |
| MAX EPOCH | 10K | 10K | 10K | 10K | 5K |
| POLICY NETWORK | [400,200,100] | [400,200,100] | [256,256,128,128] | [256, 128, 64] | [256, 128, 64] |
| CRITIC NETWORK | [400,200,100] | [400,200,100] | [256,256,128,128] | [256, 128, 64] | [256, 128, 64] |
| ACTIVATION FUNCTION | ELU | ELU | ELU | ELU | ELU |

- $r_{\text{lateral}} = v_{\text{torso},y}$ is the penalty for lateral velocity, $w_{\text{lateral}} = -1.0$

The reward is $-1$ for termination step. The action is the force applied to all joints.

*b) Humanoid Dribbling:* In this task, the robot learns to dribble along routes with random turns. The observation space is augmented with features of the ball compared with the tightrope task. For observation calculation, the global coordinate system changes with the new target route at the turning position. At each non-terminating step, the reward $r = w_v \times r_v + w_{\text{alive}} \times r_{\text{alive}} + w_{\text{dist}} \times r_{\text{dist}} + w_{\text{heading}} \times r_{\text{heading}} + w_{\text{action}} \times r_{\text{action}} + w_{\text{energy}} \times r_{\text{energy}} + w_{\text{lateral}} \times r_{\text{lateral}}$, where

- $r_v$ is the ball's forward velocity, $w_v = 2.0$;
- $r_{\text{alive}} = 1$, $w_{\text{alive}} = 0.2$;
- $r_{\text{dist}} = e^{-d}$ where $d$ is the 2d distance from torso to the ball, $w_{\text{dist}} = 0.2$;
- $r_{\text{heading}} = e_{\text{forward},x}$, where $e_{\text{forward}}$ is the basis vector of torso's $x$ axis in the global system, $w_{\text{forward}} = 1.0$;
- $r_{\text{action}}, r_{\text{energy}}$ are the same with Humanoid Tightrope
- $r_{\text{lateral}} = v_{\text{ball},y}$ is the penalty for the ball's lateral velocity, $w_{\text{lateral}} = -0.5$

The reward is $-1$ for termination step. The action is the force applied to all joints.

*c) A1 Beam:* In this task, the agent controls the quadruped robot Unitree A1 [1] to walk on a balance beam with width of $10\,\text{cm}$ following a predefined speed. Considering the width of A1 and the balance beam, it is much harder than walking on the ground. There are overall 12 motors for Unitree A1, 3 for each leg. At each non-terminating step, the reward $r = w_v \times r_v + w_{\text{alive}} \times r_{\text{alive}} + w_{\text{heading}} \times r_{\text{heading}} + w_{\text{action}} \times r_{\text{action}} + w_{\text{lateral}} \times r_{\text{lateral}}$, where

- $r_v = e^{-|v_{\text{torso},x} - v_{\text{target}}|}$ is speed tracking reward, $w_v = 1.0$;
- $r_{\text{alive}} = 1$, $w_{\text{alive}} = 1.0$;

- $r_{\text{heading}} = e_{\text{forward},x}$, where $e_{\text{forward}}$ is the basis vector of torso's $x$ axis in global coordinate system, $w_{\text{forward}} = 1.0$;
- $r_{\text{action}} = \|a\|_2^2$, where $a$ is the joints action, $w_{\text{action}} = -0.5$
- $r_{\text{lateral}} = v_{\text{torso},y}$ is penalty for lateral velocity, $w_{\text{lateral}} = -1.0$

The reward is $-1$ for termination step. The robot has a low-level joint controller. The action is the target angular position of all joints.

*d) Trifinger Move:* Trifinger [21] is a 3-finger manipulator for learning dexterity. The goal of the task is to move a cube from a random initial pose to an arbitrary 6-DoF target position and orientation. The environment is the same as that of [3], except that we remove the auxiliary penalty for finger movement, which increases the difficulty of the task. The robot has a low-level joint controller. The action is the target angular position of all joints.

*e) Ant Acrobatic:* In this task, an ant learns to do complex acrobatics (e.g. heading a pole) on a ball, which extremely challenges the ability of agents to maintain balance. The action space is 8 dimensions. At each non-terminating step, the reward $r = w_{\text{alive}} \times r_{\text{alive}} + w_{\text{action}} \times r_{\text{action}} + w_{\text{energy}} \times r_{\text{energy}}$, where

- $r_{\text{alive}} = 1$, $w_{\text{alive}} = 0.5$;
- $r_{\text{action}} = \|a\|_2^2$, where $a$ is joints action, $w_{\text{action}} = -0.005$
- $r_{\text{energy}}$ is joints power consumption, $w_{\text{energy}} = -0.05$

The reward is $-1$ for termination step. The action is the force applied to all joints.

We conclude the observation space for each task in Table II for easier reading.

TABLE II: Tasks Information

| | | HUMANOID TIGHTROPE | HUMANOID FOOTBALL | TRIFINGER MOVE | A1 BEAM | ANT ACROBATIC |
|---|---|---|---|---|---|---|
| | OBSERVATION DIMENSION | 74 | 80 | 41 | 47 | 57 |
| $o_{\text{C}}$ | TORSO | $y_{\text{TORSO}}$
$z_{\text{TORSO}}$
$v_{\text{TORSO},x}$
$v_{\text{TORSO},y}$
$v_{\text{TORSO},z}$
$\omega_{\text{TORSO},x}$
$\omega_{\text{TORSO},y}$
$\omega_{\text{TORSO},z}$
$\alpha_{\text{TORSO}}$
$\beta_{\text{TORSO}}$
$\gamma_{\text{TORSO}}$ | $y_{\text{TORSO}}$
$z_{\text{TORSO}}$
$v_{\text{TORSO},x}$
$v_{\text{TORSO},y}$
$v_{\text{TORSO},z}$
$\omega_{\text{TORSO},x}$
$\omega_{\text{TORSO},y}$
$\omega_{\text{TORSO},z}$
$\alpha_{\text{TORSO}}$
$\beta_{\text{TORSO}}$
$\gamma_{\text{TORSO}}$ | | $y_{\text{TORSO}}$
$z_{\text{TORSO}}$
$v_{\text{TORSO},x}$
$v_{\text{TORSO},y}$
$v_{\text{TORSO},z}$
$\omega_{\text{TORSO},x}$
$\omega_{\text{TORSO},y}$
$\omega_{\text{TORSO},z}$
$\alpha_{\text{TORSO}}$
$\beta_{\text{TORSO}}$
$\gamma_{\text{TORSO}}$ | $x_{\text{TORSO}}$
$y_{\text{TORSO}}$
$z_{\text{TORSO}}$
$v_{\text{TORSO},x}$
$v_{\text{TORSO},y}$
$v_{\text{TORSO},z}$
$\omega_{\text{TORSO},x}$
$\omega_{\text{TORSO},y}$
$\omega_{\text{TORSO},z}$
$\alpha_{\text{TORSO}}$
$\beta_{\text{TORSO}}$
$\gamma_{\text{TORSO}}$ |
| | TORSO JOINTS | $\theta_{\text{LOWER WAIST},x}$
$\theta_{\text{LOWER WAIST},y}$
$\theta_{\text{PELVIS},x}$
$\omega_{\text{LOWER WAIST},x}$
$\omega_{\text{LOWER WAIST},y}$
$\omega_{\text{PELVIS},x}$
$a_{\text{LOWER WAIST},x}$
$a_{\text{LOWER WAIST},y}$
$a_{\text{PELVIS},x}$ | $\theta_{\text{LOWER WAIST},x}$
$\theta_{\text{LOWER WAIST},y}$
$\theta_{\text{PELVIS},x}$
$\omega_{\text{LOWER WAIST},x}$
$\omega_{\text{LOWER WAIST},y}$
$\omega_{\text{PELVIS},x}$
$a_{\text{LOWER WAIST},x}$
$a_{\text{LOWER WAIST},y}$
$a_{\text{PELVIS},x}$ | | | |
| | EXTERNAL OBJECTS | | $x_{\text{BALL}}$
$y_{\text{BALL}}$
$z_{\text{BALL}}$
$v_{\text{BALL},x}$
$v_{\text{BALL},y}$
$v_{\text{BALL},z}$ | $x_{\text{CUBE}}$
$y_{\text{CUBE}}$
$z_{\text{CUBE}}$
$H_{\text{CUBE},x}$
$H_{\text{CUBE},y}$
$H_{\text{CUBE},z}$
$H_{\text{CUBE},w}$
$x_{\text{CUBE TARGET}}$
$y_{\text{CUBE TARGET}}$
$z_{\text{CUBE TARGET}}$
$H_{\text{CUBE TARGET},x}$
$H_{\text{CUBE TARGET},y}$
$H_{\text{CUBE TARGET},z}$
$H_{\text{CUBE TARGET},w}$ | | $x_{\text{POLE}}$
$y_{\text{POLE}}$
$z_{\text{POLE}}$
$v_{\text{POLE},x}$
$v_{\text{POLE},y}$
$v_{\text{POLE},z}$
$\omega_{\text{POLE},x}$
$\omega_{\text{POLE},y}$
$\omega_{\text{POLE},z}$
$\text{UP}_{\text{POLE},x}$
$\text{UP}_{\text{POLE},y}$
$\text{UP}_{\text{POLE},z}$
$x_{\text{BALL}}$
$y_{\text{BALL}}$
$z_{\text{BALL}}$
$v_{\text{BALL},x}$
$v_{\text{BALL},y}$
$v_{\text{BALL},z}$
$\omega_{\text{BALL},x}$
$\omega_{\text{BALL},y}$
$\omega_{\text{BALL},z}$ |
| $o_{\text{S},i}$ | LIMB JOINTS | $\theta_{\text{UPPER ARM},x}$
$\theta_{\text{UPPER ARM},z}$
$\theta_{\text{LOWER ARM},x}$
$\theta_{\text{THIGH},x}$
$\theta_{\text{THIGH},y}$
$\theta_{\text{THIGH},z}$
$\theta_{\text{KNEE},x}$
$\theta_{\text{FOOT},x}$
$\theta_{\text{FOOT},y}$
$\omega_{\text{UPPER ARM},x}$
$\omega_{\text{UPPER ARM},z}$
$\omega_{\text{LOWER ARM},x}$
$\omega_{\text{THIGH},x}$
$\omega_{\text{THIGH},y}$
$\omega_{\text{THIGH},z}$
$\omega_{\text{KNEE},x}$
$\omega_{\text{FOOT},x}$
$\omega_{\text{FOOT},y}$
$a_{\text{UPPER ARM},x}$
$a_{\text{UPPER ARM},z}$
$a_{\text{LOWER ARM},x}$
$a_{\text{THIGH},x}$
$a_{\text{THIGH},y}$
$a_{\text{THIGH},z}$
$a_{\text{KNEE},x}$
$a_{\text{FOOT},x}$
$a_{\text{FOOT},y}$ | $\theta_{\text{UPPER ARM},x}$
$\theta_{\text{UPPER ARM},z}$
$\theta_{\text{LOWER ARM},x}$
$\theta_{\text{THIGH},x}$
$\theta_{\text{THIGH},y}$
$\theta_{\text{THIGH},z}$
$\theta_{\text{KNEE},x}$
$\theta_{\text{FOOT},x}$
$\theta_{\text{FOOT},y}$
$\omega_{\text{UPPER ARM},x}$
$\omega_{\text{UPPER ARM},z}$
$\omega_{\text{LOWER ARM},x}$
$\omega_{\text{THIGH},x}$
$\omega_{\text{THIGH},y}$
$\omega_{\text{THIGH},z}$
$\omega_{\text{KNEE},x}$
$\omega_{\text{FOOT},x}$
$\omega_{\text{FOOT},y}$
$a_{\text{UPPER ARM},x}$
$a_{\text{UPPER ARM},z}$
$a_{\text{LOWER ARM},x}$
$a_{\text{THIGH},x}$
$a_{\text{THIGH},y}$
$a_{\text{THIGH},z}$
$a_{\text{KNEE},x}$
$a_{\text{FOOT},x}$
$a_{\text{FOOT},y}$ | $\theta_{\text{FINGER UPPER}}$
$\theta_{\text{FINGER MIDDLE}}$
$\theta_{\text{FINGER LOWER}}$
$\omega_{\text{FINGER UPPER}}$
$\omega_{\text{FINGER MIDDLE}}$
$\omega_{\text{FINGER LOWER}}$
$a_{\text{FINGER UPPER}}$
$a_{\text{FINGER MIDDLE}}$
$a_{\text{FINGER LOWER}}$ | $\theta_{\text{FRONT HIP}}$
$\theta_{\text{FRONT THIGH}}$
$\theta_{\text{FRONT CALF}}$
$\theta_{\text{REAR HIP}}$
$\theta_{\text{REAR THIGH}}$
$\theta_{\text{REAR CALF}}$
$\omega_{\text{FRONT HIP}}$
$\omega_{\text{FRONT THIGH}}$
$\omega_{\text{FRONT CALF}}$
$\omega_{\text{REAR HIP}}$
$\omega_{\text{REAR THIGH}}$
$\omega_{\text{REAR CALF}}$
$a_{\text{FRONT HIP}}$
$a_{\text{FRONT THIGH}}$
$a_{\text{FRONT CALF}}$
$a_{\text{REAR HIP}}$
$a_{\text{REAR THIGH}}$
$a_{\text{REAR CALF}}$ | |
| | $|\mathcal{N}|$ | 2 | 2 | 3 | 2 | 4 |
| | ACTION DIMENSION | 21 | 21 | 9 | 12 | 8 |