# OpenReview forum: "Geometric Regularity with Robot Intrinsic Symmetry in Reinforcement Learning"
_roboticsfoundation.org/RSS/2023/Workshop/Symmetry — RSS 2023 Workshop Symmetry_

### Official Review · Reviewer_9MXS · 2023-06-16
**A solid work on applying MARL to leverage symmetry in robot control**

**Rating:** 7
**Confidence:** 3

**Review:**

Summary:

This paper presents a multi-agent framework to leverage the reflectional and rotational symmetric structures in robots. The authors propose dividing the robots into several components and formulating each part as an individual agent. To leverage the symmetry property, the authors utilize parameter sharing to enable the agent and its symmetric counterparts to share the same policy network. The authors compare the proposed method with a few baselines to justify the benefits of the MARL framework and parameter sharing among symmetric components.

[Strengths]

* This paper tackles the interesting problem of how to leverage the geometric prior presented in robot structure to learn optimal control more efficiently.
* Applying MARL with parameter sharing to leverage symmetry in robot control is an interesting idea, and the proposed framework seems solid and easy to understand.
* The experiments provide sufficient evidence to support the overall framework.
* Most of the paper is well-written with good use of figures.

[Weakness]

* Equation (1) is not well-explained.
* There is a lack of comparison with baselines that also leverage the symmetric structures

---

### Official Review · Reviewer_YTcN · 2023-06-16
**Well-written paper with good results**

**Rating:** 9
**Confidence:** 4

**Review:**

Summary:
The paper explores the use of symmetric properties in robotic tasks. The authors formulate the problem under Multi Agent Reinforcement Learning framework and propose a symmetric neural network on top of the framework. The experiments cover various tasks and demonstrate a significant improvement over baselines.

Comments:
1. The authors effectively address the critical problem of how to leverage intrinsic symmetries in a system. This is particularly important for fields like locomotion where the control stability is heavily reliant on the system's state itself rather than external observations. This paper shows the potential of the idea of using symmetry with MARL framework for single robots in reinforcement learning.
2. The results are impressive. One small thing is that MASA underperforms other baselines at the very beginning in Humanoid Dribbling. It would be beneficial if the authors could provide an explanation as to why this specific environment exhibits this behavior.
3. Overall, the paper is well-written and provides clear explanations for the MARL formulation. And the paper aligns well with the topic of the workshop. However, the rotation equivariance method needs further clarification. For example, it is vague that the rotation angles in the transformation function T_i(o) on central observations is defined as "the degree of rotation is the angular distance between the robot limbs of agents i and 0". Given there are several angles, e.g., the fixed angle between the joints and the canonical base pose or the movable angle between the body angle and the limbs. My guess is the former one. An additional rotation equivariance illustration figure may help.


Conclusion:
The authors propose a novel yet generic way to leverage symmetries under the MARL framework and show impressive results in non-trivial robotic tasks. It would strengthen the paper further if sim-to-real experiments were included, such as the A1 Beam environment. Overall, this paper well demonstrates the potential of utilizing intrinsic symmetry in robot learning. I incline to accept this paper.

---

### Decision · Program_Chairs · 2023-06-24

**Decision:**

Accept

**Comment:**

Congratulations! We encourage the authors to revise the paper based on the reviewer's feedback.
Your paper will be presented as both a short presentation and a poster. Detailed instructions about the presentation format and camera-ready submission will be sent to you soon.